# Single-Cell Sequencing: An Emerging Tool for Biomarker Development in Nuclear Emergencies and Radiation Oncology

**DOI:** 10.3390/cancers17111801

**Published:** 2025-05-28

**Authors:** Jihang Yu, Md Gulam Musawwir Khan, Nada Mayassi, Bhuvnesh Kaushal, Yi Wang

**Affiliations:** 1Radiobiology and Health, Isotopes, Radiobiology & Environment Directorate (IRED), Canadian Nuclear Laboratories, Chalk River, ON K0J 1J0, Canada; 2Department of Chemistry and Biology, Faculty of Science, Toronto Metropolitan University, 350 Victoria Street, Toronto, ON M5B 2K3, Canada; 3Faculty of Medicine, The University of British Columbia, Vancouver, BC V6T 1Z3, Canada; 4Department of Biochemistry Microbiology and Immunology, Faculty of Medicine, University of Ottawa, Ottawa, ON K1H 8M5, Canada

**Keywords:** next-generation sequencing, single-cell sequencing, radiation dosimetry, radiation biomarkers, cellular heterogeneity

## Abstract

Single-cell sequencing is a state-of-the-art technology that allows scientists to investigate gene activity and DNA damage in various groups of cells. It has found broad applications in cancer research, microbiology, neurology, and immunology. Furthermore, single-cell sequencing is invaluable for identifying unknown cell types and specific genetic changes. This capability makes it a powerful tool for developing biomarkers associated with radiation exposure. These biomarkers can facilitate rapid emergency responses, improve our understanding of long-term health effects, and allow for cell-specific analysis in relation to cancer radiotherapy. As technology continues to advance, integrating various validation and analysis methods may lead to more efficient ways to measure and manage radiation exposure. In summary, single-cell sequencing holds great promise for advancing research on radiation biomarkers, particularly in the context of nuclear emergencies and radiation oncology.

## 1. Introduction

In the wake of a significant catastrophic event, such as a pandemic or a natural disaster, many individuals flood hospitals seeking medical assistance. This sudden surge can place considerable strain on medical personnel tasked with triage. In such scenarios, it is vital for Canada to be adequately prepared for potential radiological or nuclear emergencies. Effective preparedness not only ensures that triage protocols are followed but also prioritizes vulnerable populations, particularly children and young people, who are more sensitive to radiation and face greater health risks in these situations.

Exposure to ionizing radiation or other harmful chemical substances can cause chromosomal aberrations, serving as indicators of cellular damage in both normal cells and tumour tissues [1]. Since the mid-1960s, the field of biological dosimetry has evolved significantly. Among the various assays, the dicentric chromosome assay (DCA) is widely recognized as the gold standard for detecting chromosomal rearrangements in peripheral blood lymphocytes [2]. However, despite its sensitivity and specificity, DCA is relatively time-consuming, low-throughput, and labour-intensive, which limits its practicality in large-scale emergency settings [3]. To address these limitations, NGS technologies have emerged as promising alternatives for radiation biodosimetry. NGS can efficiently assess genetic abnormalities in the blood, body fluids, and other non-invasive sample types in a high-throughput manner. For example, using targeted NGS, a German research team successfully analyzed 1000 samples in less than 30 h and identified four genes associated with hematological acute radiation syndrome (HARS) [4]. Importantly, HARS-predictive genes can be detected within a critical time window of 2 h to 3 days after irradiation [5], enabling timely medical interventions. In contrast, DCA typically requires sample collection between 24 h and 4–6 weeks post-exposure for optimal results [6]. Delays in sample collection for DCA may compromise the accuracy of dose estimation and the timeliness of clinical response.

While rapid, high-throughput triage strategies are being explored for nuclear emergencies, it is anticipated that NGS technologies will play a more significant role in biomarker development rather than frontline field use for biodosimetry. Understanding radiation-induced molecular changes is essential for developing in vitro diagnostics tests, such as quantitative real-time PCR (RT-qPCR), based on validated biomarkers.

This review specifically focuses on the application of NGS technologies—particularly single-cell RNA sequencing (scRNA-seq)—in the context of radiation accidents and nuclear emergencies. Although still in its early stages, scRNA-seq is rapidly advancing and offers unprecedented resolution into cellular function and gene expression profiles at the individual cell level. This granular insight supports the identification of biomarker candidates and the development of rapid diagnostic tools, such as RT-qPCR assays. Moreover, by elucidating the molecular responses of distinct cell populations to radiation exposure, scRNA-seq can help identify cell types with increased sensitivity or resistance to radiation. These findings not only enhance emergency preparedness but also have important implications for patient management in radiation oncology.

Therefore, the objective of this review is to address a critical gap in biodosimetry whereby traditional methods are limited in speed and sensitivity and cannot capture cell-specific genomic responses or low-dose effects. It is important to note that while nuclear accidents can increase long-term cancer risk, radiation (in controlled doses) is also used therapeutically in cancer treatment. This review addresses biomarkers in both acute accidental exposure and therapeutic contexts, with an emphasis on emergency biodosimetry. To this end, we (1) summarize existing biodosimetry approaches, (2) highlight radiation-responsive biomarkers identified over the past decade, and (3) explore how scRNA-seq can complement and enhance conventional methods by providing greater sensitivity and mechanistic insight. Ultimately, our goal is to inform the development of rapid, field-deployable diagnostic assays, such as RT-qPCR, for use in radiological and nuclear emergency response.

## 2. Current Biodosimetry Methods

Biodosimetry is the assessment of biological responses to ionizing radiation exposure, aiding in the evaluation of health risks and guiding appropriate medical interventions. Methods for biodosimetry are broadly categorized into physical techniques, such as detecting radioactivity in a sample, and biological techniques, which assess radiation-induced changes in living cells or tissues [7,8,9,10]. Among biological methods, cytogenetic assays—notably the DCA—have long been considered the gold standard [11]. Moreover, emerging molecular and computational tools are revolutionizing biodosimetry capabilities by enhancing sensitivity, throughput, and applicability to low-dose or chronic exposures. Therefore, we summarize the currently available biodosimetry methods in this section.

### 2.1. Biologic Techniques

#### 2.1.1. DCA

The DCA method is widely regarded as the gold standard in biodosimetry, and was, for many years, the only validated technique available for radiation dose assessment [11]. It quantifies dicentric chromosomes—aberrant chromosomes with two centromeres—that are formed when ionizing radiation induces DNA double-strand breaks. Misrepair of these breaks can result in the fusion of chromosome fragments, producing dicentric chromosomes, which serve as a hallmark of radiation exposure [12].

In the DCA, peripheral blood lymphocytes are typically irradiated, cultured, and allowed to progress through cell division. Metaphase spreads are then prepared and analyzed using cytogenetic techniques, such as Giemsa staining or fluorescence in situ hybridization (FISH). The number of dicentric chromosomes present in the sample is then quantified and correlated with the dose estimate, for which a higher frequency of dicentric chromosomes is directly proportional to the level of damage caused by ionizing radiation [13,14].

Despite its high accuracy and reliability, the DCA has several limitations. It is primarily sensitive to doses of radiation above 100–200 mGy (milligray, a unit of absorbed dose of ionizing radiation) [15], limiting its effectiveness for detecting modest to low levels of environmental exposure. Furthermore, because dicentric chromosomes are unstable and lost over successive cell divisions, DCA is most suitable for assessing acute radiation exposure and is not applicable to chronic or long-term dose estimation.

The assay is also labour-intensive, requiring skilled personnel to manually score metaphase cells under a microscope. This low-throughput nature makes it impractical for rapid, large-scale screening, particularly during a radiological or nuclear emergency, where time and resources are known to be the bottleneck for the DCA [16,17].

#### 2.1.2. Automation in DCA

To reduce the time-consuming manual scoring process, several artificial intelligence (AI)-powered methods for DCA analysis are currently under development.

These tools aim to enhance automation and throughput. For example, Ryan’s group developed Chromosome ABerration cAlculation Software (CABAS), while Rogan’s team created an automated Desktop version (ADCI Desktop) that serves as a platform for scoring dicentric chromosomes. Shen’s group introduced the DCScore, a machine learning method-based software that employs clustering and watershed algorithms for automated detection. Wadhwa’s group proposed a deep learning model called InceptionResnetv2 for identifying dicentric chromosomes with high accuracy [18,19,20,21].

More recently, a deep learning-based automatic dose-estimation system (DLADES) was developed that integrates dicentric scoring with a well-constructed dose–response curve for dose prediction [14]. Although this system was not validated for doses above 4 Gy or below 0.5 Gy, it demonstrated potential for high-throughput assessment and improved sensitivity, including estimates for doses below 0.1 Gy, to meet the requirement for higher cell counts in conventional methods [14].

Additionally, Kim et al. applied the YOLOv5 deep neural network architecture to automate dicentric chromosomes directly from metaphase images [22]. Unlike DLADES, which focuses on dose estimation, the YOLOv5 model prioritizes the rapid identification and quantification of dicentric chromosomes without the need for dose reconstruction.

#### 2.1.3. Other Cytogenetic Biodosimetry Techniques

In addition to dicentric analysis, several other cytogenetic analysis techniques have been developed to assess radiation exposure by targeting alternative biological endpoints. These include FISH-based translocation analysis, premature chromosome condensation (PCC) analysis, and the cytokinesis-block micronucleus (CBMN) assay [11].

FISH-based translocation analysis is a robust method for retrospective biodosimetry. It detects radiation-induced chromosomal rearrangement, specifically translocations, and enables the construction of a dose–response curve based on translocation frequencies in lymphocytes [23,24,25]. While early studies raised concerns about the potential misclassification of dicentric chromosomes as translocations, this issue was resolved through the use of centromeric probes, which improved scoring accuracy [26].

PCC analysis, first introduced in 1974 using Chinese hamster ovary (CHO) cells to visualize radiation-induced chromosome damage in interphase cells, has a key advantage for evaluating non-dividing cell populations. The degree of chromosome condensation inversely correlates with the extent of radiation damage [27]. Subsequent applications in mammalian and human cells confirmed the value of PCC and demonstrated a dose–effect relationship for radiation-induced chromosome aberrations, including micronuclei formation [28,29].

Complementary to DCA, the CBMN assay is another one of the most well-validated cytogenetic methods for assessing radiation-induced chromosome damage in human lymphocytes. It offers high sensitivity, with a detection limit as low as 5 cGy [30]. Furthermore, CBMN can detect ionizing radiation-induced bystander effects and inter-individual variability in DNA damage responses, indicating it as a valuable tool in population-based radiation risk assessment [31,32].

### 2.2. Physical Techniques

Scientists now recommend changing the criteria for assessing radiation exposure. Instead of focusing solely on the estimated dose received by the individual, they suggest placing greater emphasis on the biological response of the patient as indicated by radiosensitive markers. This shift is based on advancements in biodosimetry methods developed over recent years and considers the initial decision-making necessary for triage in emergency situations [33]. The rationale behind this is that even if individuals experience the same radiation dose in a radiation disaster, their physiological responses can differ significantly. Factors such as age, sex, individual immunity, and population genetics play a significant role in these differences. Additionally, even with an accurate measurement of whole-body dose, different organs may absorb and respond to radiation in distinct ways [33]. These observations have led to a growing interest in developing tools that can provide a more personalized and biologically relevant assessment of radiation injury [34].

One notable advancement in physical biodosimetry is the use of electron paramagnetic resonance (EPR) for in vivo fingernail dosimetry, as described by Swartz et al. [33]. This technique detects radiation-induced free radicals generated in the keratin of fingernails and toenails, with the EPR signal intensity being directly proportional to the absorbed dose. The approach offers several practical advantages: it is non-invasive, requires minimal training, and allows for cross-validation across multiple limbs. Additionally, the stability of EPR signals for several weeks post-exposure enhances its utility in delayed-response scenarios. Recent improvements have also increased the detection resolution from 10 Gy to as low as 1 Gy, marking a significant milestone in in vivo dosimetry. Moreover, when combined with organ-specific biomarkers, nail EPR has the potential to provide a more comprehensive assessment of dose distribution and biological response, thereby improving clinical decision-making and resource allocation in radiation emergencies [33]. Similar applications are being explored with tooth-based EPR, although further validation is needed to confirm its triage potential [35].

This advancement in biodosimetry, from dose estimation to individualized biological assessment, sets the stage for a deeper exploration of radiation-induced biomarkers, which offer critical insights into both exposure and biological effect. Hence, Table 1 summarizes the major biodosimetry methods discussed above, including their dose sensitivity, applicability, and current limitations.

## 3. Radiation-Induced Biomarkers

Radiation-induced biomarkers and biodosimetry methods are both essential tools for assessing exposure to ionizing radiation, yet they differ in their purposes and approaches. Biodosimetry methods aim to directly estimate the absorbed radiation dose, often through biological markers like chromosomal aberrations or lymphocyte counts. In contrast, a radiation biomarker refers to a measurable biological indicator that reflects physiological, pathological, or therapeutic responses to radiation exposure [38]. Biomarkers may include genetic alterations, protein expression profiles, imaging data, blood-based markers, organ function metrics, or even electrocardiographic patterns [39].

Radiation biomarkers can provide information about the molecular and cellular consequences of radiation exposure, whereas biodosimetry methods shed light on dose quantification. Both approaches are complementary and useful for managing radiation exposure, especially in nuclear emergencies or radiological incidents, depending on the context and nature of the information required.

In addition to classical cytogenetic endpoints, such as chromosome aberrations, lymphocyte depletion, and γ-H2AX foci formation [40], advances in molecular biology have led to the identification of numerous radiation-sensitive molecules. Modern “omics” technologies, including genomics, transcriptomics, proteomics, lipidomics, and metabolomics, are being actively explored to discover novel radiation biomarkers [41].

### 3.1. Systematic Literature Review Methodology

To comprehensively review current advances, we conducted a systematic literature search in PubMed, using the keywords “biomarker”, “radiation”, and “accident”. This search yielded 360 published studies, of which 254 were screened to focus on studies from approximately the last decade. Out of these, 134 studies were further screened and reviewed according to PRISMA (Preferred Reporting Items for Systematic Reviews and Meta-Analyses) guidelines to ensure a structured and reproducible selection process. The PRISMA flowchart (Figure 1) illustrates the study selection workflow.

### 3.2. Recent Biomarker Discoveries

Originally pioneered by Olivier Guipaud in 2013, serum and plasma proteomics have become widely applied to radiation biomarker discovery [42]. For example, C-X-C motif chemokine ligand 10 (CXCL10) has emerged as a promising candidate, as its expression in mouse peripheral blood mononuclear cells (PBMCs) exposed to 1, 3, and 5 Gy of ionizing radiation showed strong correlation with ferredoxin reductase (FDXR), a well-established radiation biomarker [43]. In another study, the serum protein BPI Fold-Containing Family A Member 2 (BPIFA2) was significantly upregulated in C57BL/6J mice after radiation exposure, highlighting its potential as a novel early biomarker for nuclear incidents [44]. Additionally, a secondary increase in serum BPIFA2 showed a second peak in expression after exposure to a lethal dose (10 Gy), indicating its prognostic value for predicting fatal radiation outcomes [44]. Similarly, serum amyloid A1 (SAA1) has shown marked elevation at both protein and transcript levels in irradiated models, making it a strong early-stage biomarker for radiation-induced damage and lethality risk assessment [45]. Moreover, we compiled Table 2, which includes 49 studies highlighting biomarkers identified in recent research over the past decade. A comprehensive version can be found in Appendix A.

### 3.3. Emerging High-Throughput Platforms for Biomarker Discovery

Transcriptomics and proteomics approaches have been widely applied to identify radiation-induced biomarkers, contributing to the development of large-scale databases that offer a comprehensive understanding of radio-sensitive and responsive biomolecules. In parallel, high-throughput platforms are being developed to enable rapid and efficient biomarker discovery, with several panels of candidate biomarkers selected for validation.

For instance, the high-throughput biodosimetry test system (REDI-Dx) has been developed for investigational use in the United States [92]. Utilizing a minimally invasive blood collection technique, REDI-Dx incorporates a panel of 18 dose–response genes along with three normalizer genes and two internal controls, assessed via RNA Sanger sequencing. This system has demonstrated high sensitivity and specificity for radiation dose estimation, particularly at exposures of 2 Gy and 6 Gy [92].

In addition to transcriptomic platforms, metabolomics has emerged as a promising tool for biomarker discovery, particularly given its capacity to capture systemic metabolic dysregulation following radiation exposure [38]. Several metabolites, including carnitine (C7H15NO3) and citric acid (C6H8O7), have been consistently identified as altered in response to varying radiation doses, positioning them as potential candidates for radio-sensitive biomarkers for unexpected nuclear events [38].

Furthermore, Maan et al. reported significant dysregulation of multiple metabolic and immunological pathways following exposure to 1 Gy and ultra-high doses (7.5 Gy) of radiation [93]. These included alterations in lipid metabolism, carbohydrate metabolism, and amino acid metabolism, particularly pathways involving histidine, arginine-proline metabolism, and arginine biosynthesis. Therefore, these insights not only deepen our understanding of molecular interactions underlying radiation response but also support the integration of multi-omics approaches, combining metabolomics and transcriptomics, for the identification of robust biomarkers. Such integrative analyses have the potential to enhance triage and medical management strategies in radiation emergency scenarios [93].

## 4. Single-Cell Sequencing Technology: A New Era for Radiation Biomarker Discovery

### 4.1. Advantages of Single-Cell Sequencing over Bulk RNA Sequencing

In contrast to conventional RNA sequencing (RNA-seq), which provides an average gene expression across mixed cell populations, single-cell sequencing technologies (including genome, transcriptome, and epigenome sequencing) allow for the detection of gene signatures and differential gene expression at the resolution of individual cells. This is particularly important for addressing cellular heterogeneity within tissues or populations, which is often masked in bulk RNA-seq data. Consequently, scRNA-seq technology has rapidly emerged as a transformative tool for studying gene expression variability at the single-cell level, offering deeper insights into intra-tumour or intra-organ heterogeneity [94].

Since its initial report in 2009, high-throughput scRNA-seq technology has become an indispensable technique for studying genomic, transcriptomic, and epigenomic landscapes at the single-cell level, gaining wide adoption across fields such as developmental biology, cancer research, and immunology [95].

### 4.2. Technical Approaches for Single-Cell Sequencing

Single-cell sequencing can be applied across various molecular domains:scRNA-seq for transcriptome profiling;Single-cell DNA sequencing for genome analysis;Single-cell proteomics for global protein expression profiling in thousands of individual cells.

While scRNA-seq shares core steps with bulk RNA sequencing, the critical difference lies in the cell isolation, which includes conventional manual sorting, dilution, laser microdissection (LCM), fluorescence-activated cell sorting (FACS), and microfluidics/microplate technology [96,97,98]. Among these, microfluidics platforms greatly enhance throughput, efficiency, and accuracy, enabling the analysis of large cell populations with high fidelity [98,99].

### 4.3. scRNA-Seq in Cancer Research

Tumours exhibit extensive heterogeneity not only among cancer cell populations harbouring diverse mutations and transcriptional programs but also within the tumour microenvironment, which evolves throughout oncogenesis. This complexity poses significant challenges in biomarker discovery for diagnosis, prognosis, recurrence prediction, and therapy resistance.

scRNA-seq offers a powerful means to dissect this complexity by enabling the detection of cell type-specific gene expression patterns within tumours and surrounding stromal or immune compartments [94,100]. For instance, Gao et al. applied single-cell transcriptomics to breast cancer cells exposed to ionizing radiation, uncovering heterogeneous cellular responses and identifying potential radio-sensitive biomarkers in tumour treatment [101]. Similarly, scRNA-seq has been used to develop a risk assessment model based on epithelial cell markers in colorectal cancer [102] and reveal sex-specific gene expression in glioma-activated microglia, highlighting its value for personalized cancer research [103]. Moreover, scRNA-seq has been employed to identify biomarkers within circulating tumour cells (CTCs) from melanoma, hepatocellular carcinoma, and non-small-cell lung cancer [104,105,106].

### 4.4. Potential Applications of scRNA-Seq in Radiation Exposure Scenarios

scRNA-seq holds great promise for identifying biomarkers associated with accidental radiation exposure, nuclear incidents, and space radiation exposure. Acute high-dose radiation can trigger diverse effects on different cell populations within organs, which can be systematically profiled using scRNA-seq.

Recent studies have leveraged scRNA-seq to investigate the impact of space-like radiation conditions on human tissue. For example, researchers used engineered human bone marrow and cardiac tissue combined with single-cell transcriptomics to examine the effects of high-energy neutron and photon radiation, simulating space mission conditions. They observed a decreased proliferation of CD45+ cells, heightened inflammatory signatures, and myeloid cell infiltration and identified potential biomarkers such as MIR22HG, HMOX1, COL24A1, MIR34AHG, and PHLDA3, associated with oxidative stress, fibrosis, and senescence [107]. Additionally, scRNA-seq has been applied to analyze T lymphocyte subpopulations exposed to ex vivo radiation, revealing differential gene expression profiles among subpopulations and providing novel insights into immune responses to radiation [108].

Though still in its early stages within radiation research, scRNA-seq is poised to revolutionize biodosimetry by offering a more precise and cell-specific understanding of gene expression changes after radiation exposure, ultimately advancing personalized countermeasure development.

### 4.5. scRNA-Seq in Personalized Radiation Therapy and Immune Modulation

Radiation therapy, as the most cost-effective treatment of cancer, has been applied to more than half of cancer patients alone or in combination with surgery or chemotherapy. By profiling radiation-induced biomarkers before and after radiation, scRNA-seq can help determine individual radiation sensitivity, allowing for the design of customized treatment plans tailored to each patient’s genetic landscape. This technology also enables the characterization of the radiation-induced changes not only in tumour cells but also in the surrounding environment, such as immune cells, blood cells, and mesenchymal cells, thereby offering a holistic view of the treatment response.

Recent studies support this application. Ponthan et al. used 10× Genomics scRNA-seq to show that radiotherapy enhances the infiltration of CD8+ T cells expressing natural killer granule protein 7 (NKG7) in colorectal carcinoma, suggesting potential targets for immune modulation strategies [109]. Moreover, NFKB1 was identified via single-cell whole genome sequencing as a gene that increases the radiation sensitivity of cervical cancer cells, representing a target for precision therapy [110].

## 5. Discussion

Ionizing radiation has both harmful and beneficial effects. In uncontrolled exposures, such as nuclear or radiological incidents, it can lead to long-term health risks (e.g., cancer, cataracts, cardiovascular and neurological disorders), making rapid triage and sensitive biomarkers essential tools for effective and early intervention. In contrast, cancer radiotherapy uses carefully calibrated doses to damage tumour cell DNA while minimizing injury to healthy tissue. Single-cell sequencing unites these contexts by providing high-resolution, cell type-specific profiles of radiation-induced gene expression and DNA damage responses. This precision enables the identification of early biomarkers for emergency risk assessment and the refinement of personalized dose planning in the clinic.

Despite these clear advantages, bulk-based biomarker studies in both emergency and therapeutic settings have well-documented limitations that hinder clinical translation. Prior reviews of radiotherapy biomarkers have highlighted significant obstacles to establishing universal predictors of radiosensitivity and response, including inter-tumoral heterogeneity, variable microenvironmental influences, and limited reproducibility of bulk-based signatures [111,112]. Many candidate markers fail prospective validation because they average signals across mixed cell populations, masking rare but critical subpopulations that determine treatment outcome. Single-cell RNA and its profiling can overcome these challenges by: (1) resolving cellular heterogeneity, identifying resistant or sensitive clones that constitute a small fraction of the tumour; (2) revealing microenvironmental interactions, such as immune-tumour cell cross-talk, that modulate radiosensitivity; and (3) enabling multi-layered signatures, where transcriptomic, epigenetic, and proteomic features from the same cell can be integrated to yield more robust predictors.

Recent studies have begun to apply single-cell techniques in clinical cohorts, including pilot studies. Applying single-cell profiling in clinical cohorts has outperformed bulk assays in stratifying radiotherapy responders versus non-responders [111,113]. For instance, Li et al. developed a radio-resistance gene signature in head and neck squamous cell carcinoma by integrating transcriptomic data and patient outcomes, demonstrating improved prediction of radiotherapy response compared to traditional clinical parameters [114]. These examples underscore the potential of single-cell-derived biomarkers to resolve cellular heterogeneity and refine predictive models for both emergency biodosimetry and precision radiotherapy.

Beyond ionizing radiation alone, cells are frequently exposed to additional stressors in radiotherapy, such as engineered nanomaterials, that can further influence DNA damage responses. Chow describes nanoparticle properties such as size, shape, and surface chemistry that govern their cellular uptake and DNA interactions, leading to direct strand breaks or indirect damage via reactive oxygen species generation [115]. Nanomaterials may also interfere with DNA repair proteins, altering cellular responses to ionizing radiation. Integrating these biophysical insights with scRNA-seq would allow for the simultaneous quantification of physical lesions (e.g., via γ-H2AX imaging) and corresponding transcriptomic changes in individual cells. Such a multimodal approach can identify cell type-specific biomarkers of combined radiation–nanomaterial exposure, improving both our mechanistic understanding and the development of rapid assays for complex environmental or clinical scenarios.

## 6. Limitations and Future Prospects

Although automation has improved the efficiency of single-cell sequencing workflows, library preparation still requires at least 10 h, excluding sequencing and data analysis time [116]. Furthermore, the need for specialized equipment and stable laboratory environments poses challenges for rapid field deployment. Additionally, the analysis of single-cell sequencing data is computationally intensive and requires specialized expertise, which can be a limiting factor for rapid turnaround. Nevertheless, the unique capability of single-cell sequencing to uncover novel cellular subsets and specific gene expression signatures across diverse tissues positions it as a valuable platform for advancing biomarker discovery. It holds particular promise for the development of both rapid ex vivo diagnostic tools and deeper insights into the long-term biological effects of radiation.

Future applications of single-cell multi-omics, integrating transcriptomic, genomic, epigenomic, and proteomic data, may allow for the identification of specialized cellular responses and therapeutic targets. We propose that radiation-sensitive biomarkers identified via single-cell sequencing be compiled into a comprehensive biomarker matrix panel (see Figure 2), which could support more accurate triage and personalized medical countermeasures following radiation exposure.

## 7. Conclusions

Single-cell sequencing has revolutionized biomarker discovery by providing high-resolution insights into how individual cells respond to radiation. In nuclear emergencies, it enables the identification of early markers for rapid triage, while, in radiation oncology, these same insights can guide patient stratification and optimize therapeutic dosing. Given the urgency of such situations, single-cell sequencing is best positioned as an upstream discovery tool. The key biomarkers identified through this approach can then be translated into faster, more accessible assays, such as quantitative PCR or immunoassays, for use in routine laboratory settings and rapid triage. As the technology continues to progress, its integration with multi-omics, machine learning, and emerging fields like nanotoxicology will further deepen our understanding of complex exposures. Ultimately, this combined strategy supports both immediate response efforts after a radiation incident and personalized treatment planning in the clinic.

## Figures and Tables

**Figure 1 cancers-17-01801-f001:**
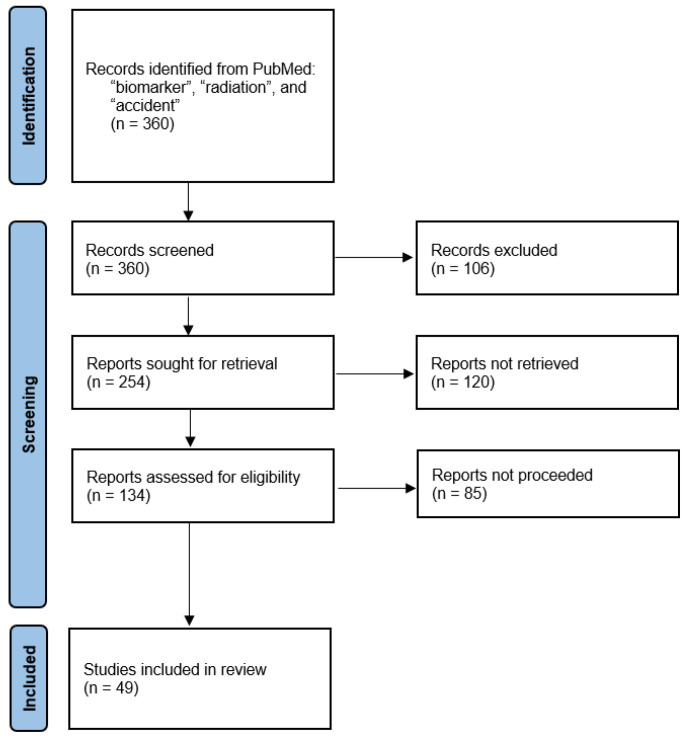
PRISMA flow diagram of the literature study. Adapted from the PRISMA template 2020 in accordance with the terms of the Creative Commons Attribution (CC BY 4.0) license.

**Figure 2 cancers-17-01801-f002:**
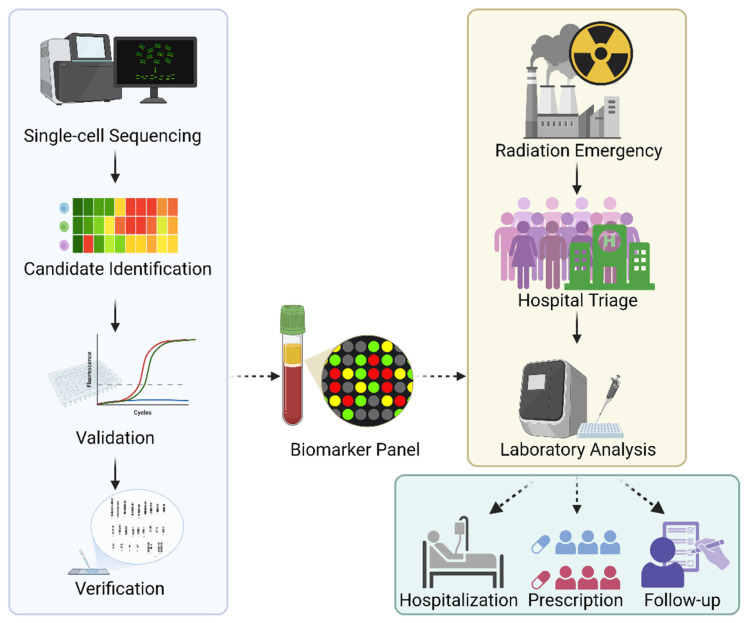
Development of the radiation-sensitive biomarker panel. Single-cell sequencing would provide biomarker candidates to be selected and validated with faster methods such as qPCR. Once the validation is completed, candidates would be measured against gold standard methods, such as DCA, for verification. The biomarker panel would benefit patients for triage and laboratory analysis in the event of a radiation emergency, which would serve not only for dosimetry and radiation protection but also for guiding treatment in hospital settings, managing prescriptions, and facilitating follow-up healthcare.

**Table 1 cancers-17-01801-t001:** Overview of current biodosimetry methods.

Method	Principle	Dose Sensitivity	Throughput	Application	Limitations	References
DCA	Dicentric chromosomes (cytogenetics)	>100–200 mGy	Low	Acute exposure, gold standard	Laborious, not for chronic/low doses	[11,12,13,15]
DCA + AI (e.g., DLADES)	Automated dicentric scoring (deep learning)	<0.1 Gy possible	High	High-throughput, emergency triage	Still under validation	[14,18,19,20,21,22]
FISH Translocation Analysis	Stable chromosomal translocations	~100 mGy	Medium	Retrospective analysis	Specialized probes needed	[23,24,25,26]
PCC Analysis	Chromosome condensation	>1 Gy	Medium	High-dose, non-dividing cells	Limited use in low-dose exposure	[27,28,29]
CBMN Assay	Micronuclei in lymphocytes	~50 mGy	Medium	Bystander effect, inter-individual dif.	Requires cytokinesis block	[30,31,32]
Nail EPR	Free radicals in keratin (physical method)	~1 Gy	High	Field deployable, non-invasive	Low-dose sensitivity is still improving	[33]
Tooth EPR	Free radicals in enamel	~1 Gy	High	Potential triage tool	Requires more validation	[35]
Biomarker-Based Assays	Gene/protein/metabolite expression changes	Variable	High	Personalized risk assessment	Standardization lacking	[36,37]

**Table 2 cancers-17-01801-t002:** Radio-sensitive biomarker candidates identified in the last decade (2012–2024).

Category	Biomarker(s)
Genome	BRAF mutation [46]; HLA-genetic predisposition [47]; CNV [48]; MD of autosomal SNPs [49]; C-3SFBP, C-7IUVU [50]
mRNAs	PTC transcriptomic signature [51]; CCNG1, PHPT1 [52]; RET/PTC1 rearrangement [46]; KDR, CEACAM8, OSM [53]; CLIP2 [54,55]; Agpat9, Plau, Prf1, S100a8 genes [56]; gene expression signatures [57,58,59]; NF-κB1, NF-κB2, Rel genes [60]; radiation-responsive “signature” genes [61]; CLIP2-PPIL3 co-expression [62]; ERCC1, ESCO2 [63]; 15 mRNAs [64]; ^131^I exposure novel biomarkers [65]; GRB7, B2M, PMAIP1 [66]; CXCL10, FDXR [43]
microRNAs	miRNA-150 [67]; miR-21 [68]; miRNA signatures [69,70]; 5-miRNA composite signature [71]
Proteins	Low proliferation index [46]; CRP [72]; CD11b+CD13+, CD29+CD13+, cell adhesion and migration [73]; chronic viral infection [74]; intracellular protein parameters [58]; SAA1 [45]; glutathione transferase, glutathione peroxidase [75]; Keratins K1 and K10 [76]; AMY1A, FLT3L, MCP1 [77]; A2m, CHGA, GPX3 [78]; γ-H2AX mean fluorescence intensity [79]; 30+ proteomic biomarkers [80]; BPIFA2 [44]; ^131^I exposure novel biomarkers [65]; BRAF/NRAS mutation, PD-L1, PD-1, P16INK4A, Ki-67 [81]; serum sSelectin-L [82]; BAX, DDB2 [83]
Metabolites	26 metabolite signals [84]; lipid metabolism (postnatal), steroid hormone metabolism (adult) [85]; urinary metabolic signatures [86]
Cells	CD4+ cells, CD4+/CD8+ ratio [53]; absolute neutrophil count, monocyte count [72]; PHA [87,88]; lectin–erythrocyte interactions [89]; telomere length [58,59,74]; TCR-CD4+, γ-H2AX+, and CyclinD1+ cell counts [90]; cellular immunity [58]; PPHA [91]

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
