# Peer review of "Single-Cell Sequencing: An Emerging Tool for Biomarker Development in Nuclear Emergencies and Radiation Oncology"

_cancers, 2025, doi:10.3390/cancers17111801_

Round 1
Reviewer 1 Report
Comments and Suggestions for Authors
Reviewer’s comments
This manuscript, "Single-Cell Sequencing: An Emerging Tool for Biomarker Development in the Even of Nuclear Emergency," is well-written, technically sound, and has great potential. However, some areas require significant revision to improve clarity and interpretability and strengthen the manuscript to reach publication standards.
Here are my comments to improve the quality of the manuscript
Major Corrections
1. The manuscript presents a comprehensive review of single-cell sequencing for radiation biomarker development, but it lacks a clear discussion on how this approach differs from existing techniques. A section comparing single-cell sequencing with traditional biodosimetry methods in terms of accuracy, sensitivity, and applicability in emergency response is needed. The authors should explicitly highlight the research gaps this review is addressing.
2. The structure of the manuscript is somewhat fragmented. A clear research question or objective should be presented in the Introduction.
3. The methodology of how studies were selected and analyzed is missing. A systematic review framework, such as PRISMA, would add rigor to the study selection process.
4. The table listing radiation biomarkers is extensive. Please add e.g., specificity, sensitivity, and validation status to make the comparison more structured. Or maybe a summary table categorizing biomarkers by effectiveness and validation level would improve readability.
5. Several sentences are long and complex. The manuscript would benefit from language editing to improve clarity and flow. Some sections are repetitive, especially in the introduction and discussion of biodosimetry methods.
Comments on the Quality of English Language
Several sentences are long and complex. The manuscript would benefit from language editing to improve clarity and flow. Some sections are repetitive, especially in the introduction and discussion of biodosimetry methods.
Reviewer 2 Report
Comments and Suggestions for Authors
Yu et al. provide a comprehensive review well describing the modern ways to measure and manage radiation exposure. The subject of the review is important and up-to date. Single cell sequencing is a modern technology which should be understood and widely applied and the review fulfills this goal. I approve the manuscript as it is
Reviewer 3 Report
Comments and Suggestions for Authors
Referee Report
This topical review discusses the use of single-cell sequencing as a tool for biomarkers in nuclear emergencies. After reviewing the manuscript, I have the following concerns:
- Title: The title does not directly relate to cancer, which is the main focus of the Journal. It appears more suitable for submission to Radiation or Cells Journals in MDPI. The authors may want to consider rephrasing it.
- Scope of DNA Damage: The authors mention DNA damage measured by single-cell sequencing, but it is unclear whether this damage is related to nuclear emergency events (e.g., nuclear accidents) or cancer treatments (e.g., radiotherapy). It is important to note that nuclear accidents can induce cancer, while radiotherapy is used to treat cancer. The authors should carefully define the scope of their review concerning these two fields, both of which involve DNA damage.
- Nanomaterials: An important area of DNA damage involves the application of nanomaterials, which is not mentioned in this manuscript. References such as Chow et al. (AIMS Biophys. 2024, 11(3), 340-369) would be useful.
- Subsections and Tables: It would be beneficial if the authors provided subsections for Sections 2-4 to help readers understand different bio-dosimetry methods, biomarkers, and single-cell sequencing technologies comprehensively. Additionally, tables summarizing all methods, biomarkers, and technologies would enhance the review. Note that Table 1 is too long (over 5 pages) and should be shortened.
- Limitations and Future Prospects: There should be a section addressing the current limitations and future prospects of the field. Moreover, a conclusion section is missing from the review.
- Tables and Diagrams: The manuscript currently includes only one table and one figure, which is insufficient. The authors are encouraged to provide more tables and diagrams.
Round 2
Reviewer 1 Report
Comments and Suggestions for Authors
Thank you for revising the manuscript as suggested
Author Response
Thank you so much for reviewing our manuscript!
Reviewer 3 Report
Comments and Suggestions for Authors
The authors have addressed all my concerns.
Author Response

(The authors gave the same response as above.)
